# Mechanical Properties of Sugar Palm (*Arenga pinnata* Wurmb. Merr)/Glass Fiber-Reinforced Poly(lactic acid) Hybrid Composites for Potential Use in Motorcycle Components

**DOI:** 10.3390/polym13183061

**Published:** 2021-09-10

**Authors:** S. F. K. Sherwani, E. S. Zainudin, S. M. Sapuan, Z. Leman, K. Abdan

**Affiliations:** 1Advanced Engineering Materials and Composites Research Centre (AEMC), Department of Mechanical and Manufacturing Engineering, Universiti Putra Malaysia, UMP, Serdang 43400, Selangor, Malaysia; faisalsherwani786@gmail.com (S.F.K.S.); zleman@upm.edu.my (Z.L.); 2Laboratory of Biocomposite Technology, Institute of Tropical Forestry and Forest Products (INTROP), Universiti Putra Malaysia, UMP, Serdang 43400, Selangor, Malaysia; khalina@upm.edu.my

**Keywords:** mechanical properties, motorcycle components, sugar palm fiber, poly(lactic) acid, hybrid composites

## Abstract

This research aims to determine the mechanical properties of sugar palm fiber (*Arenga pinnata* Wurmb. Merr) (SPF)/glass fiber (GF)-reinforced poly(lactic acid) (PLA) hybrid composites for potential use in motorcycle components. The mechanical (hardness, compressive, impact, and creep) and flammability properties of SPF/GF/PLA hybrid composites were investigated and compared to commercially available motorcycle Acrylonitrile Butadiene Styrene (ABS) plastic components. The composites were initially prepared using a Brabender Plastograph, followed by a compression molding method. This study also illustrated the tensile and flexural stress–strain curves. The results revealed that alkaline-treated SPF/GF/PLA had the highest hardness and impact strength values of 88.6 HRS and 3.10 kJ/m^2^, respectively. According to the results, both alkaline and benzoyl chloride treatments may improve the mechanical properties of SPF/GF/PLA hybrid composites, and a short-term creep test revealed that the alkaline treated SPF/GF/PLA composite displayed the least creep deformation. The findings of the horizontal UL 94 testing indicated that the alkaline-treated SPF/GF/PLA hybrid composites had good flame resistance. However, alkaline-treated SPF/GF/PLA composites are more suitable materials for motorcycle components.

## 1. Introduction

The automotive industry has traditionally placed more importance on materials with a high strength-to-weight ratio [1]. To meet this need, previous authors have worked on composite materials [2]. Many unique composite materials were developed or fabricated and used in automotive industries [3]. Both synthetic and natural polymers are used in the production of composite materials for automotive applications [4]. The use of synthetic polymers derived from petroleum sources is hazardous to the environment [5]. Motorcycles are an essential part of the automotive industry. The majority of the motorcycle’s body frame parts are composed of an Acrylonitrile Butadiene Styrene (ABS) engineering thermoplastic. ABS is a petroleum-based plastic with a high melting point, low weight, high strength, good mechanical properties, and excellent surface finish [6]. It is also not biodegradable, is not a renewable resource, and is harmful to the environment [7]. Because of the environmentally hazardous properties of ABS plastic, researchers are leaning toward biodegradable plastic to rescue the environment. Poly(lactic acid) (PLA) is an excellent choice for replacing ABS as it is a biodegradable type of plastic that is made from plant-based materials such as corn starch or sugarcane, and it is also known as ‘the green plastic’ [8,9]. Since the 2000s, efforts have been made in the field of biodegradability to commercialize PLA polymers. PLA is a recyclable, non-toxic, biobased, biodegradable, biocompatible polymer that is readily decomposable, commercially accessible, has cheap manufacturing costs, and its mechanical and physical characteristics may be modified [10]. PLA has the world’s second-highest intake rate of any bioplastic. Morao et al. [11] indicate that the manufacture and use of PLA have a very low environmental impact and they analyzed the cradle-to-gate data on the environmental footprint of PLA production. PLA’s environmental performance may be improved by improving sugarcane cultivation methods, installing better efficiency bagasse boilers at the sugar mill, reducing the use of auxiliary chemicals, and increasing the use of renewable energy in the sugar-to-PLA conversion process. Because of these properties, PLA may be used to replace petroleum polymers in a variety of biomedical, textile, plastic, 3D printing materials, and packaging applications [12,13].

Few previous works reported on replacing existing motorcycle framing parts with natural fiber-reinforced polymer composites. Brahim et al. [3] addressed textile-reinforced composites (TRC) with resin matrices that exhibited superior machining characteristics, high weight to strength ratio, and corrosion resistance. TRC material could also be used to fabricate motorcycle components [14]. Bahari et al. [15] utilized *Cocos nucifera husk* particle (CHP) or cocopeat as a filler in the polymer composite to produce motorcycle brake pads. Sukrawan et al. [16] replaced asbestos with a bamboo fiber composite for motorcycle brake pads. BMW’s new MW HP4 Race motorcycle [17] is the World’s first bike with a complete carbon fiber frame and wheels. As a result, it has a 30% weight reduction over an aluminum alloy equivalent, weighing just 171 kg when fully fueled. Rogacki et al. [18] produced a unique scooter frame made up of epoxy carbon laminate carbon fiber-reinforced plastic (CFRP) composite was utilized. The frame was designed in such manner that it improved strength and durability. Kumar et al. [19] developed the carbon fiber-reinforced epoxy composite (CFEC) that was used to make the motorcycle’s side panels and mudguards, and they compared the mechanical properties of this CFEC composite with the existed motorcycle plastic components. 

Sugar palm (*Arenga pinnata* Wurmb. Merr) fiber (SPF) is the most abundant natural fiber in Malaysia. SPF has many advantages such as low cost, biodegradability, non-toxicity, low density, and strong mechanical strength [20]. Sugar palm fibers are used to make a variety of goods. The thick, black fibers of the leaf bases are used to make rigging and brushes, as well as to caulk boats because they are resistant to wear and biological assault when exposed to seawater [21]. These fibers are also used to make brushes and brooms, roofing materials, thatching material, fishing equipment, traditional prayer headgear, and other smaller goods and handicrafts [22]. The outer section of the trunk, which is exceptionally thick and durable, is typically utilized as a material in the manufacturing of flooring and furniture, as well as boards, barrels, posts, and other items. Salwa et al. [23] conducted a life cycle assessment (LCA) analysis of a potential thermoformed SPF-reinforced sago starch composite takeaway food container, focusing on the damage assessment of the entire product system, including disposal scenarios. According to the analysis, all impact categories had an impact of less than 0.0001 DALY (disability-adjusted life year) for the Human Health damage category and less than 0.00001 species/year for the Ecosystem Damage category. However, SPF has several limitations, including excessive moisture absorption and poor modulus. The big problem is fiber–matrix adhesion, which is caused by inconsistency between hydrophilic SPF and the hydrophobic polymer matrix [24]. This problem can be solved by chemically modifying the surfaces of fibers. Several reports have been conducted that show surface improvements in sugar palm fiber after treatments [25,26,27,28].

One of the most successful chemical surface treatments for sugar palm fiber is alkaline treatment [29]. An alkaline treatment removes waxy substances and impurities, which disrupts the hydrogen bonding in the network structure, increasing the roughness of the fiber surface and allowing for better adhesion with the polymer, and also increasing the exposure of cellulose to the fiber surface [28,30]. Previous studies have shown that alkaline-treated SPF reinforced with Polypropylene [29], high impact polystyrene [31], epoxy [22], unsaturated polyester [25], and thermoplastic polyurethane (TPU) [28] composites has enhanced the mechanical properties of the entire composite. However, alkaline treatments have the added problem of fiber degradation at high concentrations, which may be addressed with a moderate chemical treatment such as benzoyl chloride. Benzoyl chloride treatment of natural fiber reduces the hydrophilic character of natural fiber and increases the adhesion with polymer matrix as well as increases the strength of composites [32]. Alkali pretreatment is used during the benzoylation process. When further treated with benzoyl chloride, these alkaline pretreated fibers cause the OH groups of the cellulose fibers to be overtaken by benzoyl groups and render them hydrophobic [33]. While many studies have been conducted on the treatment of natural fibers with benzoyl chloride, only a few studies on the treatment of SPF with benzoyl chloride are available in the literature. These studies have been on the impact of benzoylation on alkaline-treated SPF [34], and properties of benzoyl chloride-treated SPF/Glass fiber (GF) composites [35].

Hybridization of natural fiber composite with glass fiber results in enhanced mechanical properties, and this type of hybrid composite is suitable for automotive applications [24,36]. Several benefits of natural fiber-incorporated hybrid composites have previously been reported [37]. Sugar palm fiber- and glass fiber-reinforced polymers are excellent examples of hybrid composites with well-combined properties. [25,35,38]. Recently, Syafiqah et al. [39] reported that SPF/glass fiber-reinforced epoxy hybrid composite shows excellent flexural and compressive properties after treatment of SPF. Atiqah et al. [40] examined how SPF and glass fiber combined to improve the tensile and impact properties of SPF/glass fiber-reinforced TPU hybrid composites. Nurazzi et al. [25] investigated that combining GF with treated SPF improves the thermal properties of the hybrid composites.

There have been few studies which investigate the prolonged load-bearing properties of treated biocomposites. Yicheng et al. [41] investigated the creep characteristics of honeycomb-cored lightweight natural fiber-reinforced polymer sandwich panels. Durante et al. [42] studied the creep behavior of PLA reinforced with woven hemp fabric. Wang et al. [43] investigated the creep character of flax fiber-reinforced composites after treatment of flax fiber.

There have been very few studies that have determined the hardness of biobased composites. Idris et al. [44] determined the Brinell hardness value for banana peel waste/phenolic resin to replace asbestos. Ruzaidi et al. [45] studied the hardness of a palm slag filler (asbestos-free) brake pad using the Rockwell hardness test. Rashid et al. [46] investigate the Rockwell hardness characteristics of SPF/Phenolic composites and indicate that as the SPF loading in the composites increases, Rockwell hardness decreases.

Several investigations have shown that treating fiber can improve the flammability property of biocomposites. According to Bharath et al. [47], in both the UL 94 V and UL 94 HB tests, the rate of flame propagation and mass loss of treated composites were reduced, while their resistance to flame or fire was enhanced. A horizontal burning test with UL-94 was also used to examine the flammability of treated sisal fiber (SF)-reinforced recycled polypropylene (RPP) composites [48]. When compared to untreated RPP and RPP/SF composites, the results indicated a reduction in the burning rate of up to 16% and 7.42%, respectively. According to Asim et al. [49], treated pineapple leaf fiber (PALF) and kenaf fiber (KF) phenolic composites reduced fire retardancy by 50% when compared with untreated PALF and KF. 

Limited studies have been conducted to determine the best natural/green composite combination for the application of motorcycle body components. This paper aims to evaluate the mechanical properties of SPF/GF/PLA hybrid composites, to compare their properties with ABS plastic components, and also to propose biodegradable composites as an alternative to ABS plastic in the manufacturing of motorcycle components. Because SPF is hydrophilic in nature, it was subjected to treatments to reduce its hydrophilicity: alkaline treatment and benzoyl chloride treatment. Figure 1 depicts an ABS motorcycle battery housing component that was used to compare the various characteristics of the current ABS motorcycle battery housing component with the proposed SPF/GF/PLA hybrid composites. Sherwani et al. [45] showed that 70% PLA and 30% SPF exhibit good mechanical properties, primarily tensile and flexural characteristics; thus, a 70% PLA to 30% fiber ratio is considered in this research. Recently, Sherwani et al. [50] investigated the effect of three alkaline solution concentrations of 4%, 5%, and 6% for alkaline treatment, and 50 mL benzoyl chloride-treated SPFs at three different soaking durations of 10, 15, and 20 min on SPF/PLA composites and found that the best percentage is 6% alkaline and the best soak time is 15 min for improving mechanical properties of SPF/PLA composites. Because of this reason 6% alkaline and 15-min benzoyl chloride treatment is considered here. With the assistance of this research endeavor, an environmentally acceptable biodegradable material may be offered for potential use in the manufacturing of motorcycle components.

## 2. Materials and Methods

### 2.1. Materials

Sugar palm fiber was purchased from Kampung Kuala Jempol, Negeri Sembilan, Malaysia. The poly(lactic acid) (NatureWork 2003D) and benzoyl chloride with reagent plus 99% ethanol and chopped E-glass fiber were delivered by Mecha Solve Engineering, Petaling Jaya, Selangor, Malaysia. Sodium hydroxide (NaOH) pellets were supplied by Evergreen Engineering and Services, Taman Semenyih Sentral, Selangor, Malaysia. The ABS plastic motorcycle battery housing component and Havoline engine oil (SAE 20 W-40) were purchased from Yu Huat Motor Sdn Bhd., Taman Sri Serdang, Seri Kembangan, Selangor, Malaysia. Table 1 described the various properties of SPF, GF, and PLA.

### 2.2. Preparation of Sugar Palm Fibers

A bundle of SPFs was crushed with a PrimeHub Tokai 500 DM crusher machine. Dry SPF was graded to a length of 10 mm to 15 mm. The fibers were then washed with water multiple times to eliminate any contaminants on the SPF. The SPF was left outdoors for 24 h before being dried in an air-circulating oven at 60 °C. 

### 2.3. Chemical Treatments

#### 2.3.1. Alkaline Treatment

Sugar palm fibers were alkaline-treated to remove surface contaminants as well as hemicelluloses within the fibers [24]. Fifty grams of sugar palm fibers was immersed in a 1000 mL (6% *w*/*v* NaOH) alkaline solution for 60 min at 25 °C. These were then soaked in a glacial acetic acid solution until they reached a neutral pH, rinsed with distilled water, dried in an oven (4 Factory 20 Shanghai Lichen Bangxi Instrument Technology Co., Ltd., Shanghai, China) for 24 h at 60 °C, and last packed in zipper plastic storage bags.

#### 2.3.2. Benzoyl Chloride Treatment

Fifty grams of SPF was submerged in an 18% NaOH solution for 30 min before being washed twice with water. These SPFs were suspended in a 10% NaOH solution and rapidly agitated in 50 mL benzoyl chloride for 15 min. According to Izwan [34], 15 min of soaking time for benzoylation treatment yielded the highest SPF characteristics for use as reinforcement in composites. After that, once again SPFs were rinsed with water, filtered, and dried at 25 °C. Sugar palm fibers were commonly soaked in ethanol for an hour before being washed, filtered, and dried for 24 h in a 60 °C oven.

### 2.4. Fabrication of SP/GF-Reinforced PLA Hybrid Composites

The biocomposites were developed using a melting compounding technique and a hot press molding method. Sugar palm fiber (10–15 mm size), chopped E glass fiber, and pellet PLA were dried at 60 °C in an electric oven for 48 h. Two sets of (30/70) wt.% of SPF/PLA and GF/PLA non-hybrid composites and two sets of (15/15/70) wt.% of treated SPF/GF/PLA hybrid composites were developed in the form of the composite plate. The formulation is mentioned in Table 2. To ensure uniform mixing, sugar palm and chopped E-glass fibers reinforced with poly(lactic acid) were combined for 10 min at 160 °C at a rate of 50 rpm in a Brabender Plastograph (co-rotating twin-screw extruder, Model 815651 Brabender^®^, Duisburg, Germany). These samples were then crushed with the use of a crusher machine. For hot press molding, a compressive molding Techno Vation, Selangor, Malaysia machine type 40 tons was used. These samples were preheated for 7 min at 170 °C before being fully pressed for 6 min. Three vent cycles occurred. Finally, the cold-pressed time at 25 °C was 6 min. All of the tests were carried out at room temperature at about 25 ± 3 °C. It should be mentioned that, because of the short duration of the creep tests, the effects of aging and secondary crystallization were not taken into account in this research.

## 3. Characterization of SPF/PLA Composites

### 3.1. Rainwater Absorption

For the rainwater absorption test, three samples of size 10 × 10 × 3 mm^3^ were prepared [54]. *W*_0_ and *W*_1_ values were determined for the samples before and after being submerged in rainwater for 48 h at room temperature. The average of the three samples was used to get the final result. Equation (1) was used to calculate rainwater absorption %.
(1)$$\mathrm{Rainwater}\;\mathrm{absorption},\;\%=\frac{W_{1}-W_{0}}{W_{0}}\;\times \;100$$
where *W*_0_ = the sample’s weight before immersion in rainwater and *W*_1_= the sample’s weight after 7 days in rainwater.

### 3.2. Engine Oil Absorption

Three samples of size 10 × 10 × 3 mm^3^ were prepared [46] for the oil absorption test. Initially, the samples were oven-dried for 24 h at 60 °C, then promptly weighed to the closest 0.0001 g. Each specimen’s initial weight was recorded as ‘$W_{i}$’ before soaking in SAE 20 W-40 engine oil. After soaking in engine oil at temperatures ranging from 25 to 32 °C, At the same time, the samples were drawn from the oil and wiped clean using a dry towel and weighed again as ‘$W_{e}$’. The difference in initial and end weights for each specimen was then used to calculate the oil absorption rate. Equation (2) was used to calculate the oil absorption rate %.
(2)$$\mathrm{oil}\;\mathrm{absorption}\;\mathrm{rate}\;\%=\frac{W_{e}-W_{i}}{W_{i}}\;\times \;100$$

### 3.3. Hardness Test

The Rockwell hardness test was used to test the composites’ indentation resistance (model Mitutoyo ATK 600, Mitutoyo Corp., Kanagawa, Japan). According to ASTM D785 [46], a scale S with a steel ball of diameter (1/2)” was used. Then, a 100 kg load was applied. The digital scale was used to read Rockwell hardness (HRS) values. For each specimen, at least five readings were taken to establish an average as a final reading.

### 3.4. Tensile Test

A tensile test was performed using an Instron3366 universal testing machine (UTM, University Ave Norwood, MA, USA) in accordance with ASTM Standard D638–10 [55]. Using a 5 KN load cell, the gauge length of the non-hybrid and hybrid composites was 80 mm, and the crosshead velocity was 2 mm/min. Five samples were tested, each measuring 150 mm × 25 mm × 3 mm in the shape of a rectangular composite plate. The final result was determined by taking the average of the five samples. 

### 3.5. Flexural Test

Flexural properties of non-hybrid and hybrid composites were evaluated using an Instron 3365 dual column tabletop UTM with a span length of 50 mm and a crosshead speed of 12 mm/min in accordance with the ASTM D790 (3-point bending) standard [56]. The composite plate yielded five composite samples measuring 127 mm × 12.7 mm × 3 mm. The final result was determined by taking the average of the five samples.

### 3.6. Impact Test

The dimensions of the five samples were according to ASTM D256–10 [57], 65 mm × 15 mm × 3 mm, which were separated from the composite plates for the Izod impact test. For the impact investigation, Rayran RR/IMT/178 Izod impact testers (Capacity 5.5J, Ray-Ran test equipment Ltd., Nuneaton, UK) were used. Five identical samples of each type of composite were tightly positioned vertically and struck in the center of the instruments with the help of a pendulum at the force of 10 J. The impact had 2.75 J of energy and a velocity of 3.46 m/s; the averages of the five samples were used to get the final results. 

### 3.7. Compressive Test

Compressive testing was performed in accordance with ASTM D695–15 [58]. The compressive strength testing was carried out using an Instron3366 UTM with a load cell of 100 kN. The compressive specimens were loaded at a rate of 2 mm/min. The specimens have sizes of 7.94 mm (L), 1.27 mm (W), and 3.2 mm (T). The findings for each loading were derived from a mean of five replicates.

### 3.8. Flammability UL-94

The flammability of the various composites was determined using the horizontal burning test by UL-94 in accordance with the ASTM D635 standard [59]. The sample dimension size was 125 mm × 13 mm × 3 mm. The sample was held horizontally, and a natural gas-fueled flame was utilized to fire the sample end. The time it took the flame to go from the first mark (25 mm from the end) to the second mark (100 mm from the end) has been recorded. The flame was applied to the bottom of the sample for 10 s, then replaced and repeated for another 10 s.

As soon as the sample was changed, the time of burning and glowing was recorded. 

### 3.9. Creep Test

Creep characteristics were determined by evaluating specimens in flexure in accordance with ASTM D 2990 [60]. Specimens were made and put on a rack with the same dimensions as the flexural test specimens. Composites were loaded for 300 min in three-point bending, with stresses expressed as 20% and 30% of the ultimate flexural strength of the composites [61]. All composites were evaluated using identical dead loads; composite results are presented as a percentage of ultimate flexural strength.

### 3.10. Statistical Analysis 

SPSS software was used to perform an analysis of variance (ANOVA, IBM SPSS Statistic version 23, Armonk, NY, USA) on the obtained experimental results. Duncan’s test was employed to conduct a mean comparison at a 0.05 level of significance (*p* ≤ 0.05).

## 4. Results and Discussion

### 4.1. Rainwater Absorption

Figure 2 represents the rainwater absorption against time for ABS and various composites after 7 days of testing. The capacity for water absorption of natural fiber composites depends on the density, the existence of voids, and the bond between the fiber and the matrix [62]. Water is trapped in the void as a result of these factors, increasing the composite weight. The ABS absorbs the least rainwater, while the SPF/PLA composite absorbs most of the rainwater on the first day. Table 3 represents the rainwater absorption % value for various composites. 

Sherwani et al. [63] addressed that SPF consists of holocellulose, which enhanced water absorption. Because SP fiber cells have a high concentration of hydroxyl groups, this hydrogen bond was broken when H_2_ O contacted the fiber, and the hydroxyl group formed a new hydrogen bond with H_2_ O molecules. The GF/PLA also absorbs very little rainwater as GF is hydrophobic in nature. The rainwater absorption percentage for SPF/PLA ranges from 5.91% to 15.11%, which decreases for USP/GF/PLA ranges from 4% to 7.5%. After alkaline and benzoyl chloride treatment of sugar palm fiber, this absorption percentage drops to approximately 3% to 8%. Recently, Verma et al. [64] report that water absorption capacity was decreased by 23% in alkali-treated kenaf fiber composites, whereas glass fiber hybridization reduced water uptake by 35% when compared to untreated kenaf fiber composites. The elimination of waxy substance and hemicellulose after alkaline treatment reduces the voids in the composite and enhances the adhesion between the fiber and the matrix [29], while Benzoyl chloride treatment aids in the reduction of SPF’s hydrophilic character and improved the fiber–matrix adhesion, increasing composite strength while decreasing water absorption [39]. Among all of the aforementioned composites, ASP/GF/PLA is the best proposed composite for fabricating motorcycle components in terms of rainwater absorption. 

### 4.2. Engine Oil Absorption

Figure 3 depicts the oil absorption capacities of ABS and various composites for 7 days of testing. The graph demonstrates that the percentage of oil absorption ranged from 0% to 3%. On the first day of testing, ABS absorbs the most oil, and by contrast GF/PLA absorbs nothing due to glass fiber’s hydrophobic nature. As the number of days increases, this % oil absorption value increases. The oil absorption of the SPF/PLA composite ranges from 1.32% to 2.74%, whereas the alkaline-treated SPF (ASP/GF/PLA composite) ranges from 0.54% to 0.82%. Table 4 represents the oil absorption % value for various composites. The high percentage of oil absorption % in SPF/PLA can be explained as the natural fibers are hydrophilic, oil absorption increases as the natural fiber content increases in the composites. Swelling of the fiber creates micro-cracks in the brittle matrix, which may function as a microchannel for oil transport [46]. The percentage of oil absorption in composites is also affected by the percentage of voids present. The oil absorption capacity decreases as the proportion of voids or gaps decreases [65]. After alkaline and benzoyl chloride treatments, the voids % reduces and the adhesion between SPF, GF, and PLA improves, resulting in fewer voids or gaps in composites and reduces oil absorption capacity [66]. Oil absorption ranges from 0% to 3%, while water absorption ranges from 0% to 15%. This discrepancy may be because water is less viscous than engine oil.

### 4.3. Hardness Testing

Figure 4 shows the hardness resistance values of ABS and various composites. The non-hybrid SPF/PLA composite hardness resistance value of 65.10 HRS was greater than the ABS of 57.33 HRS. According to Al Maadeed et al. [67], glass fiber-reinforced composites have a higher hardness than other composites, the hardness value of the GF/PLA composite improved to 85.27 HRS after incorporating glass fiber with PLA. Mixing alkaline-treated SPF/GF-reinforced PLA enhanced the hardness even further. Among all materials, the alkaline-treated SPF/GF/PLA had the greatest hardness value of 88.6 HRS. This might explain the excellent distribution and dispersion of alkaline-treated SPF, which results in strong interfacial bonding between fiber and PLA matrix. A positive particle dispersion can improve particle–matrix interaction and, as a result, increase the composite’s resistance to indentation. [46]. 

According to Shaniba [68], the chemical treatment of filler enhances the hardness of composites; the addition of additional filler to the rubber matrix decreases the plasticity of composites while increasing their rigidity. The BSP/GF/PLA composite shows the lowest value of hardness 46.57 HRS. We predict that after benzoyl chloride treatment, the SPF becomes soft, and on incorporation of GF/PLA, the matrix also becomes ductile, which decreased the hardness of the entire composite [69]. Table 5 represents Hardness HRS for various composites. The above hardness testing demonstrates that the proposed ASP/GF/PLA composite is more suited for motorcycle component fabrication than ABS component. 

### 4.4. Tensile Testing

The tensile properties in terms of the tensile stress–strain curve of ABS and various composites are shown in Figure 5. Curves demonstrate that the tensile stress increases proportionately to the strain, as predicted by Hooke’s law until it reaches the proportional limit. The proportionate limitations vary according to the composites. Because of the brittle nature of PLA, all of the PLA composites failed after the first crack. It is believed that the crack begins as the tensile load was applied to the samples. 

The nature of ABS plastic was ductile as the graph indicated that it became extended for up to maximum limit but was able to tolerate minimum tensile stress among all the composites. GF/PLA composites had the maximum slope in the curves. This shows that after incorporating the glass fiber, the tensile strength of the PLA composite was increased significantly. Mukhtar et al. [29] also report on the improvement of tensile properties in SPF/polypropylene with the incorporation of glass fiber. On comparing non-hybrid SPF/PLA, USP/GF/PLA, ASP/GF/PLA, and BSP/GF/PLA hybrid composites, the maximum slope curve was exhibited by the ASP/GF/PLA hybrid composite. The alkaline-treated SPF exhibited a good hydrophobicity and form a strong adhesive bonding with the PLA matrix, because of this treatment and incorporation of glass fiber, the tensile strength resistance increases. The alkaline treatment removes lignin and the waxy coating, resulting in a decrease in SPF thickness, because of which the adhesion between the fibers and the matrix was improved. According to Radzi et al. [28], the 6% alkaline-treated SPF results in the highest tensile strength of Roselle fiber/SPF hybrid composites. 

The tensile curve of benzoyl chloride-treated SPF has a lower slope as compared to USP/GF/PLA composites, which means BC surface treatment causes slip-hardening pullout behavior, which reduces crack-bridging efficiency owing to fiber breakage at small crack apertures. [70]. A similar tensile curve was showed by Hassan et al. [71] for oil palm empty fruit bunch fiber-reinforced epoxy composite. It has also been found that combining sugar palm fiber with glass fiber enhances the overall mechanical properties of the composite. Furthermore, the tensile results clearly show the advantages of the alkaline chemical treatments on the hybrid composites. The blue curve of tensile stress-strain shows that the tensile characteristics of the proposed ASP/GF/PLA composite are superior to those of the ABS component, indicating that these biodegradable composites are appropriate for motorcycle components fabrication.

### 4.5. Flexural Testing 

The flexural properties in terms of flexural stress–flexural strain curve of ABS and various composites are shown in Figure 6. Variations in flexural stress–flexural strain values exist based on the composite sample. It is believed that the fracture starts on the tension side of the specimen and gradually propagates in the horizontal direction of the specimen. The slope of the flexural stress–strain curves was also shown to increase for alkaline treated SPF. The alkaline-treated SPF had the highest slope in the curves, followed by the ASP/GF/PLA hybrid composites. The alkaline treatment removes lignin, hemicellulose, waxy layer, provide the rough fiber surface resulting in a decrease in SPF diameter. The interfacial bonding between SPF and PLA matrix was also enhanced [28]. This clearly shows that alkaline treatment improves the flexural properties of SPF/GF/PLA hybrid composites. Due to the high strength and modulus of GF mixed randomly with SPF, it tends to withstand the maximum load before bending failure. This is because the existence of sufficient fibers allows adequate stress transfer between the reinforced SPF and the PLA matrix [39]. The flexural property of SPF improved significantly after alkaline treatment as compared to untreated USPF/GF/PLA and SPF/PLA. Furthermore, it is evident from Figure 6 that hybridization improves flexural characteristics. This result is similar to the SPF/GF-reinforced polypropylene examined by Mukhtar et al. [29]. Another study found that the fiber treatment had a significant impact on the flexural strength value [24]. Alkaline treatment of fiber can minimize cell wall thickening, resulting in better fiber–matrix adhesion.

The results reveal that BSP/GF/PLA also shows a high slope curve of the flexural stress–strain curve. The BC treatment reduces the diameter of the SPF as well as removes the lignin and wax layer on the fiber. Because of the availability of benzene rings in the benzoyl group linked to the fibers, the compatibility between fiber and matrix improved. Safri et al. [39] also observed an increase in the flexural property after BC treatment of SPF/GF/epoxy hybrid composites. The flexural property of SPF decreases after benzoyl chloride treatment. The USP/GF/PLA hybrid composite could not withstand the bending stress, indicating that the untreated SPF is the result of poor wettability with the PLA matrix. The ABS plastic has a ductile character; however, it cannot withstand high flexural stress but extend for a long duration. The flexural properties of the proposed ASP/GF/PLA composite outperform those of the ABS component, showing that these biocomposites are suitable for motorcycle component fabrication work. 

### 4.6. Compressive Testing 

Figure 7 shows the compressive load–extension curve of the ABS and SPF/GF/PLA composites subjected to a compressive load. The graph exhibits nonlinear behavior, which is consistent with the previous study’s results [39]. The ultimate compressive strength of the composite is mostly determined by the matrix strength. On the other hand, the stiffness of the composite is strongly dependent on its reinforcement. Composite samples under compression often fail owing to shear crippling or kinking at the point of load application, as well as a combination of shear and compression. [72]. This assertion is supported by the compressive modulus results shown in Figure 7, which show that the GF/PLA composites have the highest compressive modulus. The shear–compression failure mode is the most common for composite specimens. When the fiber’s ultimate compressive stress is exceeded, the matrix/fiber interface may fracture in shear, leading to ultimate failure. After the ultimate failure, ABS, SPF/PLA, and GF/PLA curves indicate the progressive breakdown of the fibers. During compression testing, composites will undergo matrix yield followed by fiber micro-buckling, local fiber micro-buckling with an elastic matrix, shear failure, and pure fiber compressive failure [73]. However, on comparing composites made from USP/GF/PLA, ASP/GF/PLA, and BSP/GF/PLA. It has been observed that fiber treatment has no significant effect on the compressive properties of hybrid composites. BSP/GF/PLA composites have more compressive load-bearing capacity than the ASP/GF/PLA composites. Here, for compressive test, benzoyl chloride treatment is more effective than alkaline treatment. To eliminate some of the hydrogen bonds in the cellulose chains, SPF was pretreated with NaOH. This increases the surface reactivity of the fiber to benzoylation. By decreasing the hydrophilicity of the SPF with benzoylation, the compressive properties were enhanced. This increases the SPF’s compatibility with the hydrophobic matrix.

### 4.7. Impact Testing

Figure 8 shows impact strength kJ/m^2^ for ABS and various composites. The maximum impact strength value of 3.10 kJ/m^2^ was achieved by the ASP/GF/PLA (alkaline-treated SPF) hybrid composite due to the removal of hemicellulose, lignin, and pectin; wax generation of moisture resistance; and the creation of rough fiber surface after alkaline treatment, which improved the adhesion between treated fiber and matrix. The impact strength was directly related to the toughness of a whole composite. Fibers have an important role in impact resistance, with the combined effect of both fibers improving impact strength. Because of the brittle nature of glass fiber, the failure mechanism was GF fracture rather than GF pull-out, as the wt.% of glass fiber content was increased during hybridization. As a result of the inclusion of wt.% glass fiber content, the composite can sustain a high-speed impact load. Due to the interaction of fibers, energy dissipated in the frictional sliding of one fiber with the other. Moreover, impact strength was increased after hybridization, which increased the stress capabilities. The impact strength of an untreated SPF/PLA composite was 2.09 kJ/m^2^, which improved to 3.07 kJ/m^2^ for a non-hybrid GF/PLA composite. According to Uma Devi et al. [74], with the addition of GF, the impact strength of short pineapple fiber/GF/polyester hybrid composites improved. The impact strength of the BSP/GF/PLA composite was also increased to 2.78 kJ/m^2^, while USP/GF/PLA exhibits an impact strength value of 2.7 kJ/m^2^ only. This may be attributed to excellent interlocking between the treated fiber and the matrix, which allowed for maximal energy absorption and prevented fracture propagation, therefore improving impact properties. Thiruchitrambalam et al. [75] claimed a 12% increase in impact strength for palmyra palm leaf stalk fiber-polyester composites after BC treatment. Swain et al. [76] also revealed that after BC treatment of jute fiber, the impact strength improved for the development of jute/epoxy composites. Table 5 represents the impact strength kJ/m^2^ for various composites. Impact testing demonstrates that the proposed ASP/GF/PLA composites have greater impact strength than ABS, indicating that these biocomposites are acceptable for motorcycle component fabrication work.

### 4.8. Creep Properties 

To better understand the creep behavior of ABS and various composites, 3-point bend creep deformation tests were performed. The stresses applied to the composites were approximately 20% and 30% of the ultimate strength of the composites. Figure 9 shows Creep deformation (%) versus Time (minutes) for ABS, SPF/PLA, GF/PLA, USP/GF/PLA, ASP/GF/PLA, and BSP/GF/PLA composites. The average rates of creep deformation for short-term creep testing were determined at stress levels of 20% and 30% for each type of composite. Higher creep stress levels resulted in more initial deflection and progressive deformation. The maximum creep deformation was shown by ABS at 20% and 30% of ultimate stress level for 300 min. The decreasing creep deformation trend is ABS > GF/PLA > SPF/PLA > USP/GF/PLA > BSP/GF/PLA > ASP/GF/PLA. ASP/GF/PLA exhibits lower creep deformation relative to the BSP/GF/PLA composite at comparable stress levels, which reflects the moduli of these composites. This test indicates that after alkaline and benzoyl chloride treatments, the adhesion between SPF/GF and PLA increased, allowing these biobased composites to withstand a steady load for a longer period. This shows how PLA composite creep life can be improved by alkaline treatment of sugar palm fiber and hybridization with glass fiber because the ASP/GF/PLA composite revealed a minimum creep strain % of 0.1 to 0.2. At the creep stress levels investigated and the short duration of the creep tests, this high stiffness was most likely a significant contributor to the amount of deformation. It should be mentioned that for this study, creep deformation was tested for composites kept at a steady temperature and moisture levels. Similar short-term creep results were reported by Miller et al. [61] for poly (β-hydroxybutyrate)-co-(β-hydroxyvalerate) (PHBV)-reinforced bidirectional woven natural fiber textiles.

Zhang et al. [77] also reported that fiber treatment can improve the creep behavior of short carbon fiber-reinforced polyetherimide composites. Wang et al. [43] studied the creep life for 16 days and assessed the effect of alkali treatment for flax fiber-reinforced composites, predicting that the creep behavior improved by 31.56% when compared to the untreated composite.

### 4.9. Flammability UL 94 Tests

The flammability of a composite is dependent not only on the matrix polymer and the types of fiber, but also on interfacial bonding between the two [78]. Table 6 shows the detailed results of the UL-94 tests for ABS and other various composites. All three composites—ABS, SPF/PLA, and GF/PLA—burned up to the 100 mm marked. The flame retardancy was improved after SP fiber treatment because the specimens were not completely burnt in the horizontal UL-94 test. According to Asim et al. [49], the reduction in fire retardancy is related to the cellulose content in natural fibers after treatment. The cellulosic content was exposed after the alkaline treatment, which decreased the fire-retardant properties. Therefore, among all of these composites, the ASP/GF/PLA hybrid composite showed the highest stability against flame, making it acceptable for motorcycle component applications. Figure 10 shows the partially burned ASP/GF/PLA and BSP/GF/PLA composites.

Thermal analyses, morphological investigations, and other physical properties will be the focus of future research. More research is required before these composites can be used to fabricate motorcycle parts in real-world applications.

## 5. Conclusions

The research carried out in this paper aimed to develop a hybrid composite to propose a potential use in motorcycle components. Based on the obtained test results, the following conclusions can be drawn:
The alkaline treatment and benzoyl chloride treatment of SPF can improve the absorption resistance of biobased composites, as determined by rainwater and oil absorption tests.It has also been revealed that combining SPF with GF enhances the overall mechanical properties of the composite. Furthermore, the tensile, flexural, hardness, and impact findings clearly show the advantages of the alkaline treatments on the SPF/GF/PLA hybrid composites.Compressive properties were decreased after both alkaline and benzoyl chloride treatments of SPF for SPF/GF/PLA hybrid composites.A short-term creep test indicates that the minimum creep deformation was exhibited by the ASP/GF/PLA composite.Through the horizontal UL 94 testing, the outcome revealed that the ASP/GF/PLA hybrid composite has the highest flame resistance.

The overall research concludes that the novel ASP/GF/PLA composite is best suited for making biobased motorcycle components to preserve the environment due to its excellent mechanical properties.

## Figures and Tables

**Figure 1 polymers-13-03061-f001:**
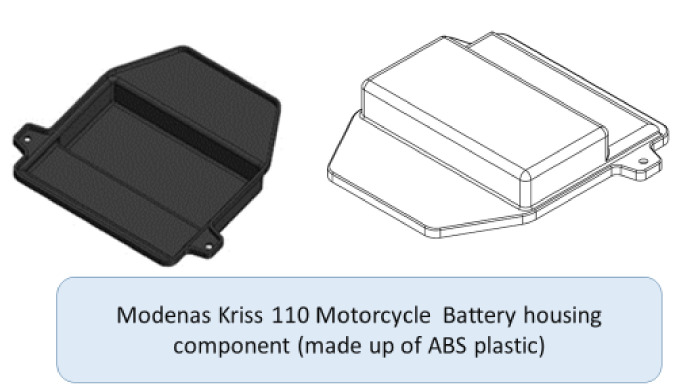
ABS battery housing component of a motorcycle.

**Figure 2 polymers-13-03061-f002:**
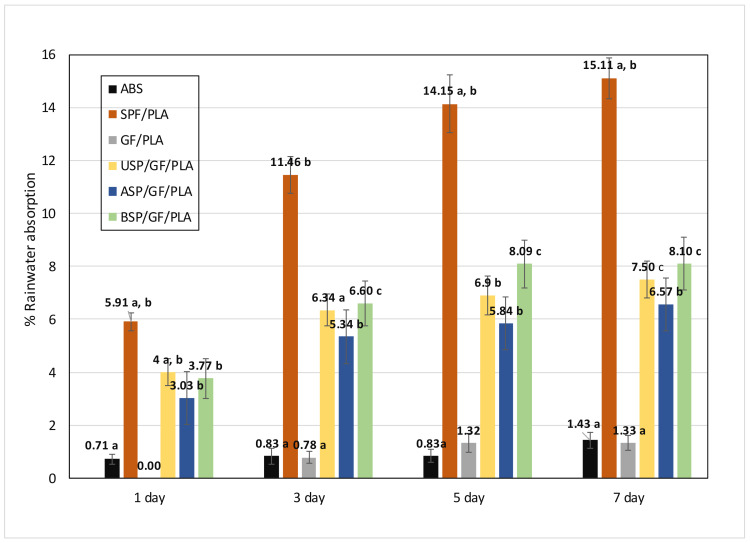
Rainwater absorption % against days after 7 days of testing. (Values with different letters in the same column are significantly different (*p* < 0.05)).

**Figure 3 polymers-13-03061-f003:**
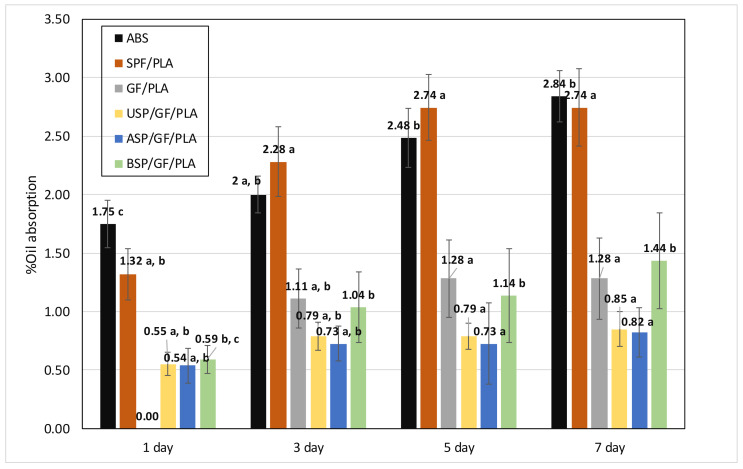
Oil absorption % against days after 7 days of testing. (Values with different letters in the same column are significantly different (*p* < 0.05)).

**Figure 4 polymers-13-03061-f004:**
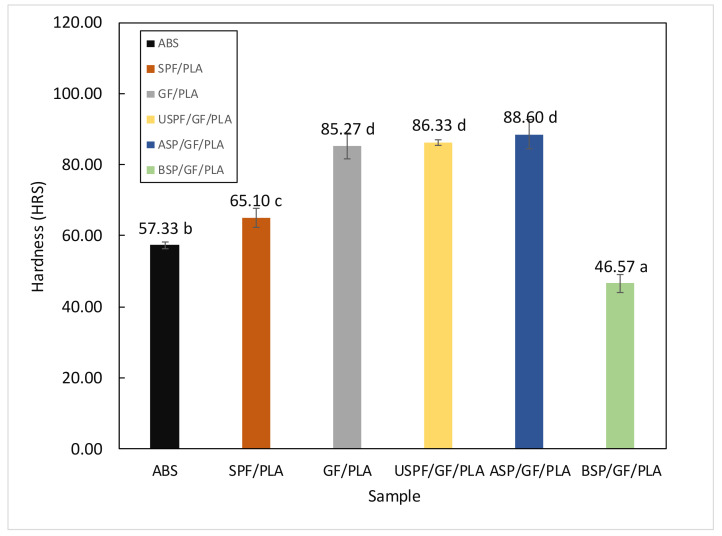
Hardness resistance value for various composites. (Values with different letters in the same column are significantly different (*p* < 0.05)).

**Figure 5 polymers-13-03061-f005:**
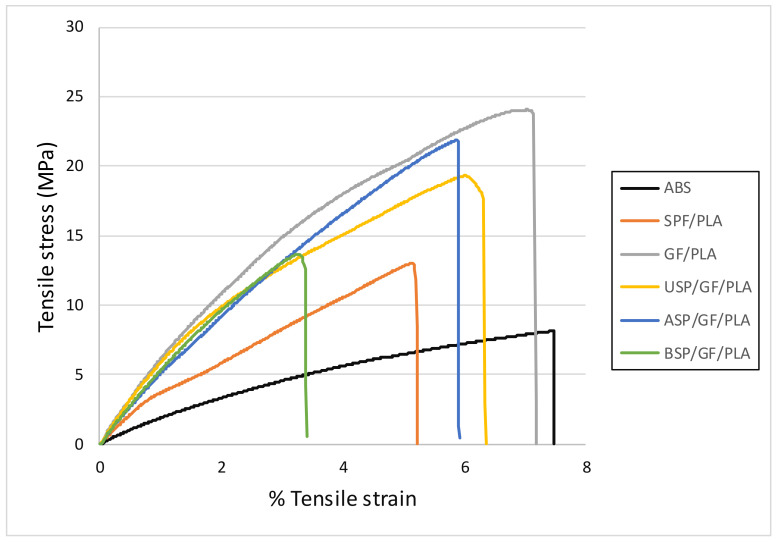
Tensile stress (MPa) versus % tensile strain for various composites.

**Figure 6 polymers-13-03061-f006:**
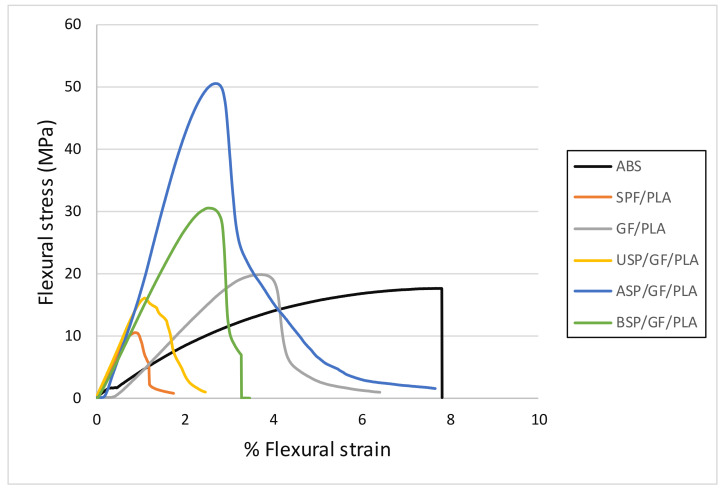
Flexural stress (N) versus % flexural strain for various composites.

**Figure 7 polymers-13-03061-f007:**
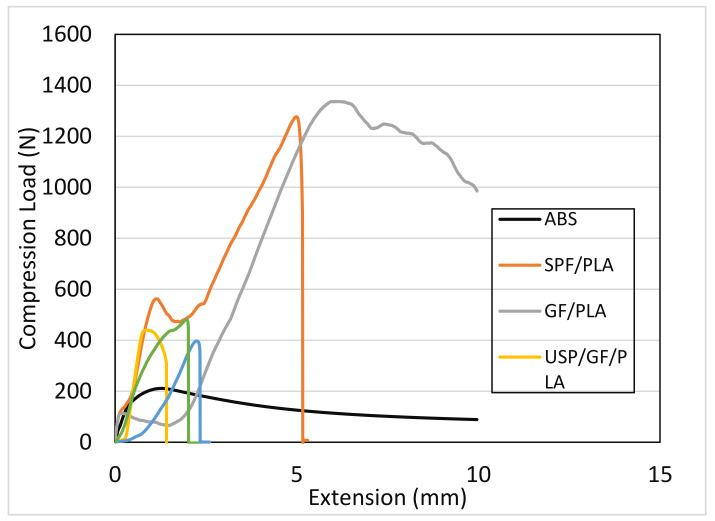
Compressive load (N) versus extension (mm) for various composites.

**Figure 8 polymers-13-03061-f008:**
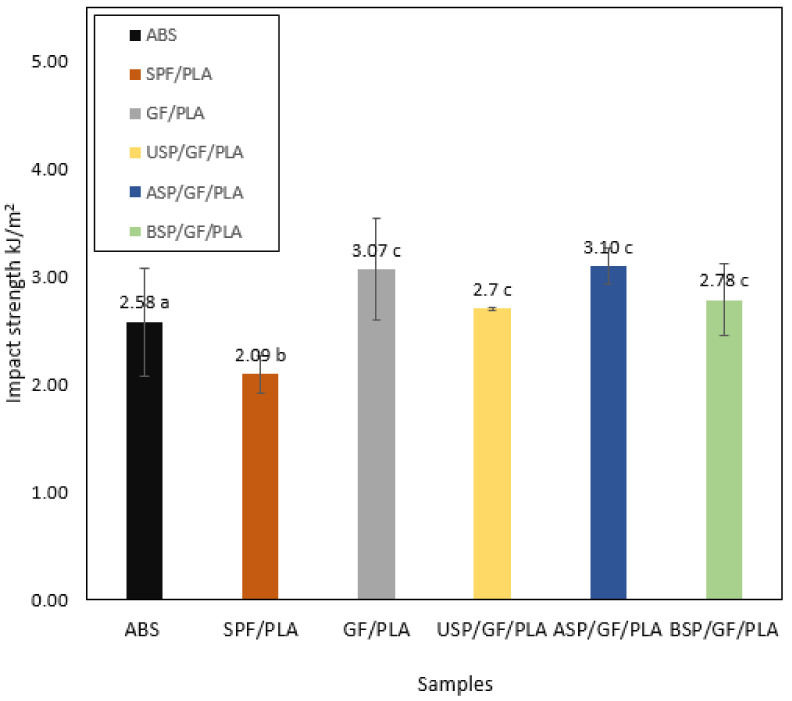
Impact strength kJ/m^2^ for various composites. (Values with different letters in the same column are significantly different (*p* < 0.05)).

**Figure 9 polymers-13-03061-f009:**
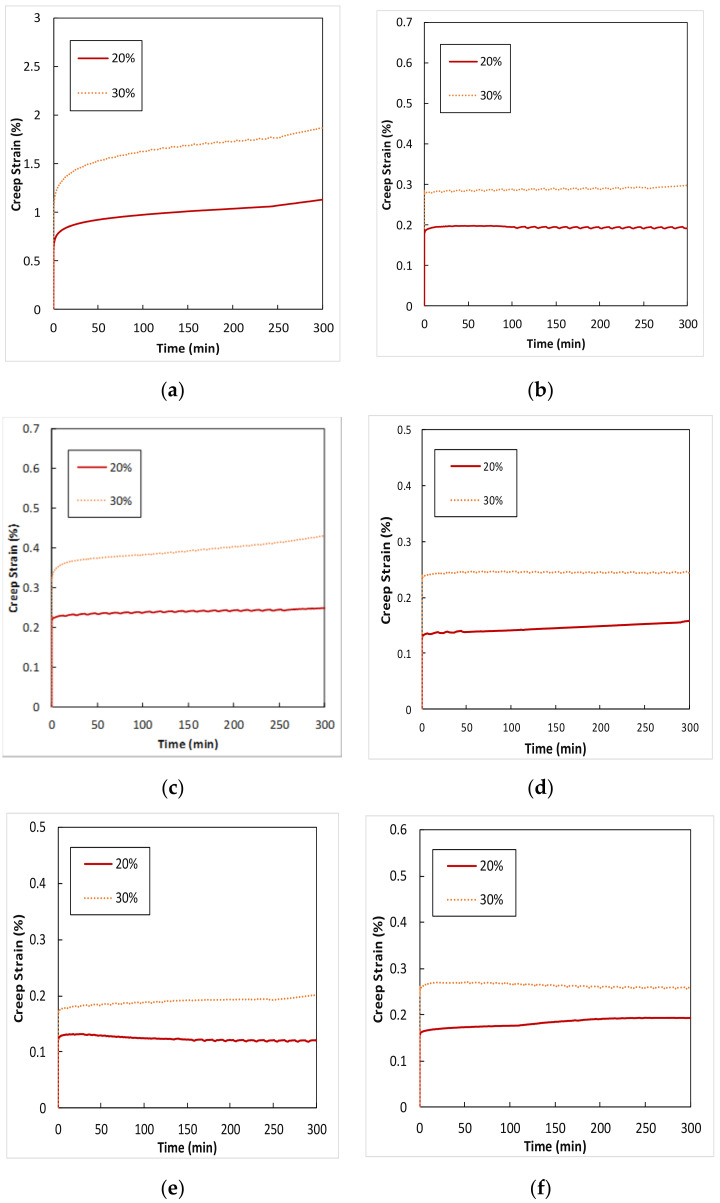
Creep deformation (%) versus time (minutes) for (**a**) ABS, (**b**) SPF/PLA, (**c**) GF/PLA, (**d**) USP/GF/PLA, (**e**) ASP/GF/PLA, and (**f**) BSP/GF/PLA composites.

**Figure 10 polymers-13-03061-f010:**
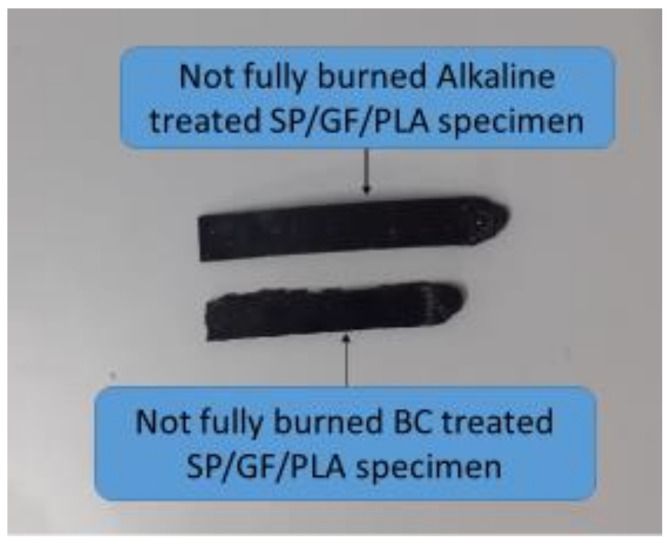
Partially burned ASP/GF/PLA and BSP/GF/PLA composites.

**Table 1 polymers-13-03061-t001:** Properties of SPF, GF, and PLA [(C_3_H_4_O_2_)_n_].

Properties	SPF [20]	GF (E-Glass) [51]	PLA [52]
Average diameter (mm)	0.5 mm	10 μm	-
Density	1.2–1.3 gm/cm^3^	2.53 gm/cm^3^	1.25–1.30 gm/cm^3^
Tensile strength	15.5 MPa	1.7–3.7 GPa (at 25 °C)	37–52 MPa
Tensile Modulus	4.189 GPa	76 GPa (single fiber)	3.5 GPa
Flexural Modulus	-	-	4 GPa
Melting Point	-	1140 °C	173 °C
Glass Transition Temperature	-		60 °C
Biodegradable	Yes	No	Yes
Decomposition temperature	228–312 °C [53]	-	-

**Table 2 polymers-13-03061-t002:** Formulation of ABS plastic, hybrid, and non-hybrid composites.

No. of Samples	Matrix	Reinforcement		
	PLA (wt.%)	SPF		GF (wt.%)
		Treatment	(wt.%)	
ABS	**Original Motorcycle Battery Housing Parts**			
SPF/PLA	70	-	30	-
GF/PLA	70	-	-	30
USP/GF/PLA	70	-	15	15
ASP/GF/PLA	70	6% NaOH	15	15
BSP/GF/PLA	70	15 min BC	15	15

BC: Benzoyl chloride treatment.; USP: Untreated SPF.

**Table 3 polymers-13-03061-t003:** Rainwater absorption % value for various composites.

Samples	Day 1	Day 3	Day 5	Day 7
ABS	0.71 ± 0.7 ^a^	0.83 ± 0.52 ^a^	0.83 ± 0.52 ^a^	1.43 ± 0.85 ^a^
SPF/PLA	5.91 ± 0.35 ^a,b^	11.46 ± 0.7 ^b^	14.15 ± 1.1 ^a,b^	15.30 ± 0.77 ^a,b^
GF/PLA	0.00	0.78 ± 0.24 ^a^	1.32 ± 0.36 ^b^	1.33 ± 0.29 ^a^
USP/GF/PLA	4 ± 0.5 ^a,b^	6.34 ± 0.6 ^a^	6.9 ± 0.75 ^b^	7.5 ± 0.7 ^c^
ASP/GF/PLA	3.73 ± 0.99 ^a,b^	5.34 ± 1.0 ^b^	5.84 ± 1.10 ^b^	6.57 ± 0.95 ^b^
BSP/GF/PLA	3.77 ± 0.74 ^b^	6.6 ± 0.85 ^c^	8.09 ± 0.9 ^c^	8.10 ± 0.99 ^c^

Values with different letters in the same column are significantly different (*p* < 0.05).

**Table 4 polymers-13-03061-t004:** Oil absorption % value for various composites.

Samples	Day 1	Day 3	Day 5	Day 7
ABS	1.75 ± 0.20 ^c^	2.00 ± 0.16 ^a,b^	2.48 ± 0.25 ^b^	2.84 ± 0.22 ^b^
SPF/PLA	1.32 ± 0.22 ^a,b^	2.28 ± 0.30 ^a^	2.74 ± 0.28 ^a^	2.74 ±0.33 ^a^
GF/PLA	0.00	1.11 ± 0.25 ^a,b^	1.28 ± 0.33 ^a^	1.28 ± 0.35 ^a^
USP/GF/PLA	0.55 ± 0.10 ^a,b^	0.79 ± 0.12 ^a,b^	0.79 ± 0.11 ^a^	0.85 ± 0.15 ^a^
ASP/GF/PLA	0.54 ± 0.15 ^a,b^	0.73 ± 0.15 ^a,b^	0.73 ± 0.35 ^a^	0.82 ± 0.21 ^a^
BSP/GF/PLA	0.59 ± 0.12 ^b,c^	1.04 ± 0.30 ^b^	1.14 ± 0.40 ^b^	1.44 ± 0.41 ^b^

Values with different letters in the same column are significantly different (*p* < 0.05).

**Table 5 polymers-13-03061-t005:** Hardness HRS and Impact strength kJ/m^2^ for various composites.

Sample	Hardness HRS	Impact Strength (kJ/m^2^)
ABS	57.33 ± 0.98 ^b^	1.37 ± 0.50 ^a^
SPF/PLA	65.10 ± 2.7 ^c^	2.09 ± 0.17 ^b^
GF/PLA	85.27 ± 3.75 ^d^	3.07 ± 0.47 ^c^
USP/GF/PLA	86.33 ± 0.76 ^d^	2.70 ± 0.02 ^c^
ASP/GF/PLA	88.60 ± 4.11 ^d^	3.10 ± 0.17 ^c^
BSP/GF/PLA	46.57 ± 2.55 ^a^	2.78 ± 0.33 ^c^

Values with different letters in the same column are significantly different (*p* < 0.05).

**Table 6 polymers-13-03061-t006:** UL 94 test results.

Sample	Rate of Burning (mm/min.)	V-2 Ranking Due to the Burning Drops Has Ignited the Cotton.	Burning Behavior
ABS	32.84	V-2	Fully burned, Fast dripping, high flame, produce high smoke, and left residue.
SPF/PLA	28.84	V-2	Fully burned, slow dripping, low flame, produce smoke, and left residue.
GF/PLA	52.44	V-2	Fully burned, Fast dripping, high flame, produce smoke, and left less residue.
USPF/GF/PLA	40.81	V-2	Partially burned, Moderate dripping, low flame, produce smoke, and left residue
ASP/GF/PLA	36.36	V-2	Partially burned, very slow dripping, very low flame, produce smoke, and left residue.
BSP/GF/PLA	39.06	V-2	Partially burned, very slow dripping, very low flame, produce smoke, and left residue.

## Data Availability

The data that support the findings of this study are available from the corresponding author, upon reasonable request.

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
