# Peer review of "Mechanical Properties of Sugar Palm (Arenga pinnata Wurmb. Merr)/Glass Fiber-Reinforced Poly(lactic acid) Hybrid Composites for Potential Use in Motorcycle Components"

_polymers, 2021, doi:10.3390/polym13183061_

Round 1

Reviewer 1 Report

This study revealed some good characteristics of different sugar palm fiber/ glass fiber/ PLA composites. The sugar palm fibers were treated by benzoyl chloride or alkaline resulting in distinguished properties, including rainwater/ oil absorption, tensile, hardness, impact, compressive, flexural, creep, and flammability. The data and discussion could be valuable and add certain contribution to the field of renewable composites. However, the writing should be improved. It is suggested that the authors should also provide data of the control materials (untreated SPF/ GF/ PLA samples). Surface morphology and properties of the modified sugar palm fibers should be provided as well. The interfacial interactions of the fibers with polymer matrix should be studied and discussed in the manuscript.   

Reviewer 2 Report

I recommend publication of this manuscript but after a major revision for more clarification and improvement. The comments are as follows:

  • Abstract, the comparison must be based on the ultimate strength, not the ultimate load in tensile and flexural test results.
  • All symbols and abbreviations used in either the abstract or the rest of the paper must be explained only once from their first appearance.
  • Table 1, where is the unit of tensile Modulus of SPF (4.189)?
  • Sections 4.1 and 4.2. It must be explained and discussed well why the resistance to water absorption differs from that of oil for each composite?
  • Figures 1 to 7, the color of the symbol of each type of composites must be the same in all of these figures.
  • Figure 4 must be converted to a stress-strain curve.
  • Figure 5 must be converted to flexural stress- flexural strain curve; see Fig. 2, and Eqs. 4 & 5 in Ref. [46].
  • The flexural and tensile strengths of each composite must be compared and discussed.
  • Tangent Modulus of Elasticity measured from the Flexural and tensile tests must be compared and discussed.
  • Conclusions should be reduced.

Reviewer 3 Report

This work studies the mechanical properties of sugar palm fiber (Arenga Pinnata Wurmb. Merr) (SPF)/ glass fiber (GF) reinforced poly(lactic acid) (PLA) hybrid composites for potential use in motorcycle components. This work is within the scope of the journal.

Language is overall good, but changes are required throughout the manuscript. For example, in the last paragraph of the introduction section "Few research have been conducted", "This aim of this paper is to compares" and elsewhere.

Why is the research focusing on motorcycle parts? What is the special characteristic that the properties of these composites fit for this use, needs to be highlighted in a better way in the manuscript.

Although several tests have been conducted in this work, further research is required for use of these materials in real applications, this should be mentioned and analyzed in the manuscript in the discussion section, which is now missing.

What is the improvement, apart from the environmental impact, for the change from ABS, the existing parts are manufactured to PLA composites, should be explained, it terms of mechanical properties, cost, time and production difficulty.

Why these specific additives were selected for this application should be explained. Why adding Sugar palm on the PLA is suitable for motorcycle parts.

The applications and use of Sugar palm should also be presented. Why it is important and how it is used in industrial applications, should be explained.

The references in the study need to be enriched. Although the literature review is adequate regarding the industrial applications of polymers, the research related on the subject of this work on polymers and their composites is missing. How they are produced, types of composites used, etc. Literature review should focus also on the materials technology and literature findings on the subject. This should help present the contribution to the field of this work, which is now totally missing from the introduction section and should be clearly presented.

What is done, how and what was found, should be more analytically presented in the last paragraph of the introduction section.

lines 3-8 of the introduction section: please add references

The type of the materials used should be mentioned. Are the properties shown in Table 1 for the specific materials used? Polymers properties from different vendors, can significantly differ.

In table 1 ABS are mentioned on the caption but are missing from the table.

Section 2.2 "crusher machine", which one should be mentioned.

Why Alkaline treatment and Benzoyl Chloride treatment are required should be mentioned.

How these loadings were selected on the composites should be explained, why 70% PLA composites were developed and why composites with only one matrix material concentration. What is the purpose of using only SPF as an additive on the PLA and since the authors developed PLA/GF composites, their properties should be evaluated with literature.

Section 3 "Characteristic of SPF/PLA composites" title is confusing, probably "Characterization" is more appropriate

Section 3.4 what type of specimen was prepared, according to the standard? Also, why 2mm/min was the speed test, which is not according to the standard?

Section 3.6 were the tests conducted according to a standard?

What ABS specimens were tested, with their results shown in figure 4? Why the force load extension graphs are shown and not the stress strain ones? Are these results comparable? Same for the remaining mechanical tests.

The manuscript reads like a technical report with no actual discussion about the results, why this outcome is expected or not, no evaluation with literature and no analysis or even comparison between the results determined in the experiments.

Authors state in the title that this research is for motorcycle parts, then they present a number of experiments they have conducted, but how these are related to motorcycle parts is missing after the introduction section of the manuscript. Which parts specifically and why? Why the mechanical behavior of the materials they develop fits for this use should be again analyzed and explained.

Authors mention about and ANOVA statistical analysis, but it is missing from the manuscript.

Round 2

Reviewer 1 Report

The authors have revised and addressed some of the concerns. The manuscript can be accepted to publish in Polymers after these minor corrections:

  1. Please correct the errors and format of Table 1.
  2. Please give definition of USP/GF/PLA sample 
  3. Please delete "Plot area" in Figure 2.
  4. On page 17 "Impact testing demonstrates that the proposed ABS/GF/PLA composites have greater impact strength than ABS components...". Please correct ABS/GF/PLA composites, should be SP/GF/PLA composites. What are the "ABS components"? It is better to use ABS sample/ copolymer. Please change it accordingly in the manuscript. 

Reviewer 2 Report

The authors have successfully addressed all my comments.  Therefore, I recommend the publication of this manuscript.

Reviewer 3 Report

The revised version of the manuscript is significantly improved in its technical aspects. All of the comments of this reviewer have been adequately replied and corresponding amendments have been made in the revised version of the manuscript. So, manuscript can be published in its current form.